# Controlled (Co)Polymerization of Methacrylates Using a Novel Symmetrical Trithiocarbonate RAFT Agent Bearing Diphenylmethyl Groups

**DOI:** 10.3390/molecules26154618

**Published:** 2021-07-30

**Authors:** Alvaro Leonel Robles Grana, Hortensia Maldonado-Textle, José Román Torres-Lubián, Claude St Thomas, Ramón Díaz de León, José Luis Olivares-Romero, Luis Valencia, Francisco Javier Enríquez-Medrano

**Affiliations:** 1Centro de Investigación en Química Aplicada, Enrique Reyna Hermosillo, No. 140, Col. San José de los Cerritos, Saltillo 25294, Mexico; robles.leonel.m19@ciqa.edu.mx (A.L.R.G.); hortensia.maldonado@ciqa.edu.mx (H.M.-T.); roman.torres@ciqa.edu.mx (J.R.T.-L.); ramon.diazdeleon@ciqa.edu.mx (R.D.d.L.); 2CONACyT-Centro de Investigación en Química Aplicada, Enrique Reyna Hermosillo, No. 140, Col. San José de los Cerritos, Saltillo 25294, Mexico; claude.stthomas@ciqa.edu.mx; 3Red de Estudios Moleculares Avanzados, Clúster Científico y Tecnológico BioMimic, Campus III, Instituto de Ecología, Xalapa 91073, Mexico; jose.olivares@inecol.mx; 4Biofiber Tech Sweden AB, Norrsken Hourse, Birger Jarlsgatan 57 C, SE-113 56 Stockholm, Sweden; luis.valencia@biofibertech.com

**Keywords:** RAFT polymerization, methacrylates, block copolymers

## Abstract

Herein, we report a novel type of symmetrical trithiocarbonate chain transfer agent (CTA) based diphenylmethyl as R groups. The utilization of this CTA in the Reversible Addition-Fragmentation chain Transfer (RAFT) process reveals an efficient control in the polymerization of methacrylic monomers and the preparation of block copolymers. The latter are obtained by the (co)polymerization of styrene or butyl acrylate using a functionalized macro-CTA polymethyl methacrylate (PMMA) previously synthesized. Data show low molecular weight dispersity values (Đ < 1.5) particularly in the polymerization of methacrylic monomers. Considering a typical RAFT mechanism, the leaving groups (R) from the fragmentation of CTA should be able to re-initiate the polymerization (formation of growth chains) allowing an efficient control of the process. Nevertheless, in the case of the polymerization of MMA in the presence of this symmetrical CTA, the polymerization process displays an atypical behavior that requires high [initiator]/[CTA] molar ratios for accessing predictable molecular weights without affecting the Đ. Some evidence suggests that this does not completely behave as a common RAFT agent as it is not completely consumed during the polymerization reaction, and it needs atypical high molar ratios [initiator]/[CTA] to be closer to the predicted molecular weight without affecting the Đ. This work demonstrates that MMA and other methacrylic monomers can be polymerized in a controlled way, and with “living” characteristics, using certain symmetrical trithiocarbonates.

## 1. Introduction

Since its early report at the end of the 1990s [1], the reversible addition-fragmentation chain transfer (RAFT) polymerization gained the attention of the scientific community. This arises as a result of the capacity to polymerize vinyl monomers and allowing the preparation of polymers with complex architectures, such as block copolymers, star copolymers and dendrimers [2,3,4,5] adapting to different reactions media [6,7,8]. The popularity of RAFT has increased impressively throughout the years. Indeed, this technique counts thousands of reports in the scientific literature [9,10,11]. This technique is part of the reversible-deactivation radical polymerization (RDRP) processes [12,13], which allows polymer scientists to prepare advanced materials with specific properties. RAFT is essentially a radical polymerization technique that provides “living” characteristics to conventional or free radical polymerization, and thus, allows the control of kinetics and molecular weight characteristics [14,15]. A successful control in the RAFT polymerization depends on the selection of an appropriate chain transfer agent (CTA, Z-(C=S)-S-R), which in turn, depends on the type of monomer to polymerize, as well as the reaction conditions [9]. The CTAs contain both Z and R groups in their structures and both play a critical role in the development (well-control and “living” behavior) of the polymerization reactions. Many investigations have reported the effect of the Z and/or R groups [11], leading to different families of CTAs, such as: dithioesters [16,17], trithiocarbonates [18], dithiocarbamates [19] and xanthates [20]. Furthermore, some switchable RAFT agents based on dithiocarbamate have been prepared and demonstrated the ability to synthesize well-defined polymers from the use of several vinyl monomers. Despite these efforts, no CTA could be considered as a universal RAFT agent, as the polymerization reaction depends on the kind of monomers and conditions processes [11]. In this sense, vinyl monomers can be classified in those in which the polymerizable double bond is conjugated to another double bond, including those of aromatic rings (i.e., styrene), carbonyl (i.e., methyl methacrylate), or nitrile groups (i.e., acrylonitrile) and that are typically controlled by dithioesthers or trithiocarbonates. On the other hand, there are those in which oxygen (i.e., vinyl acetate), nitrogen (i.e., N-vinylpyrrolidone) or halogen (i.e., vinyl chloride) are adjacent to the polymerizable double bond and that are typically controlled by dithiocarbamates or xanthates.

Among the CTAs used in the RAFT polymerization, and excluding the dithiobenzoates, the trithiocarbonates (Z = S-alkyl) are classified amongst the most reactive and mainly used to polymerize monomers like styrene (St) [21], methyl methacrylate (MMA), and/or butyl acrylate (BuA) [5], among many others [22,23,24]. While the Z group of the CTA mainly governs the stability of the intermediate radical (formed during the addition-fragmentation), the R group plays an important role in the control of the polymerization process. The efficiency of the radical leaving groups (R) to re-initiate the polymerization depends on different parameters, such as steric hindrance, radical stability, polar factor, and their chemical nature [25]. Generally, secondary, or tertiary radicals from β-scission of asymmetrical CTA have demonstrated an excellent control on the polymerization of styrenes, acrylates, acrylamides, and methacrylates monomers. On the other hand, symmetrical CTAs (trithiocarbonates) have typically shown poor control in the polymerization of methacrylate monomers, even when the chemical nature of the leaving radical group is secondary or tertiary. Therefore, we postulate that symmetrical trithiocarbonates with adequate R groups have not been designed or synthesized for the controlled RAFT polymerization of methacrylic monomers. Therefore, a controlled polymerization of methacrylates demands to this date the use of asymmetrical RAFT agents with R groups able to eject a tertiary or very specific secondary radicals [26,27], such as cumyl dithiobenzoate or cyanoisopropyl dithiobenzoate [1].

Despite the extensive effort in this field, the polymerization of methacrylic monomers using symmetrical trithiocarbonates (R-S-(C=S)-S-R) as RAFT agents remains unfruitful with a weak control in the polymeric chains. For example, S,S’-bis(α,α’-dimethylacetic acid) trithiocarbonate or the corresponding methyl ester were reported for bulk, solution, or microemulsion MMA polymerizations with a modest control [28,29,30,31], such as the values reported by Lai et al. which are *M_n_*_,SEC_ = 5.7 Kg/mol, *M_n_*_,Th_ ≈ 3.3 Kg/mol and Ð = 1.72 [30]. Moreover, the use of symmetrical trithiocarbonates with typical primary or secondary R groups, like benzyl or (α-methyl)benzyl groups, cannot be found in the literature for polymerizations of MMA and other common methacrylic monomers, as these reactions result in uncontrolled polymerizations with very broad molecular weight distributions and unpredictable molecular weights.

In this work, we introduce the di(diphenylmethyl) trithiocarbonate (see Figure 1a) as a novel RAFT agent that can specifically carry out, in a controlled manner, the polymerization of the MMA and other methacrylic monomers. Based on our results, we observed that the presence of diphenylmethyl groups (R) in this symmetrical trithiocarbonate are efficient for controlled polymerization of MMA. The evidence found suggests that this compound does not behave like a common RAFT agent since it requires atypical high molar ratios [initiator]/[CTA] for accessing to predictable molecular weights. Due to its structural characteristics, the diphenylmethyl is a good leaving group that appears to be ejected, according to RAFT mechanism, as a very stable (by resonance towards two phenyl groups) secondary radical that would generally cause inhibition or retardation in a RAFT polymerization, but in this case, it can both re-initiate and control efficiently the MMA polymerization. Furthermore, we demonstrated the potential of the functionalized PMMA as macro-CTA to prepare block copolymers by chain extension through the reaction with St or BuA. These results provide key insights with respect to the synthesis of a new family of trithiocarbonate based CTAs (including asymmetrical ones) which could have great potential and diverse applications.

## 2. Results and Discussion

### 2.1. Chain Transfer Agents

RAFT polymerization is a powerful technique used for tailoring the architecture, chain length distribution, and composition of polymers. The efficiency of the RAFT process mainly depends on the chemical nature of the CTA (or RAFT agent) and the monomer(s). Therefore, the synthesis of novel CTA represents an opportunity to achieve functionalized RAFT polymers with a specific structure. A new symmetrical RAFT agent di(diphenylmethyl) trithiocarbonate here named CTA-1 was prepared. The synthesis of CTA-1 was carried out in a one-pot reaction using the resin amberlyst-A26(OH), carbon disulfide (as reagent and solvent), and the corresponding alkyl halide (Figure 1a). The reaction yield was 92% and the chemical structure of CTA-1 was validated by ^1^H and ^13^C NMR (Figure 1b,c). The ^1^H NMR spectrum shows the characteristic signals of the aromatic protons at 7.3 ppm as a multiplet. The proton attached to the tertiary carbon that bears two phenyl substituents and neighboring a sulfur atom is observed as a signal (singlet) at 6.5 ppm. The integration of both signals match with the number of protons expected in the CTA-1 structure, that is, an integral value of 1 for the CH signal and an integral value of approximately 10 for the aromatic protons. On the other hand, the ^13^C NMR spectrum shows a signal at 218 ppm that is characteristic of the C=S group. Moreover, the presence of aromatic carbons (125 to 140 ppm), and the carbon signal of the CH with phenyl substituents and neighboring sulfur at 58 ppm are confirmed.

Furthermore, CTA-1 was used in the bulk polymerizations of methacrylic monomers (MMA, EMA, and GMA), St, and BuA at 60 °C for a predetermined time. In addition, one of the most cited RAFT agents the di(benzyl) trithiocarbonate, known as CTA-2, was prepared, and characterized in a similar way to CTA-1. Further, it was used as a reference for comparing the polymerization results obtained from both CTA.

### 2.2. Methyl Methacrylate

It is well-known that symmetrical CTAs lead to poor control in the polymerization of MMA. Table 1 shows the results derived from the polymerization of MMA in the presence of both CTAs. Here, a different molar ratio [CTA]/[initiator] was used for the polymerization of MMA with CTA-1 and as a consequence the conversion of monomer consumption varies from 54 to 91%. To our delight, these result are in accordance with the increase of [AIBN]. Indeed, the Ð of PMMA is 1.3, no matter the ratio [CTA]/[initiator] used.

Unprecedentedly, the *M_n_*_,SEC_ of PMMA obtained at a higher [AIBN] is a little closer to the *M_n_*_,Th_ than those acquired at low [AIBN]. The MMA experiments proceeded nearly to full conversion (91% for Entry 1) after 15 h of polymerization using the CTA-1 (Figure 2a), this conversion is slightly higher when compared to Entry 2 that uses the conventional CTA-2, both experiments were carried out at a typical molar ratio of RAFT polymerizations. Moreover, a nearly linear increase of *M_n_* as a function of the conversion using CTA-1 is showed in Figure 2b. It suggests the “living” behavior of this polymerization, which possibly is the fact that the conversion rate could be described as a first-order kinetics relation (Figure 2a, inset). Figure 2c shows the expected complete shift of the SEC traces towards a higher molecular weight region as the MMA polymerization in the presence of CTA-1 progresses, conserving the different chromatograms an apparent monodispersity. Such monomodality, as well as the low Ð values of around 1.3 for these PMMA samples (aliquots at different times for Entry 1), agreeing the control of the polymerization. The MMA polymerization control using CTA-2 (Entry 2) was rather poor, clearly shown by the exorbitant molecular weight value found in this sample, *M_n_*_,SEC_ of 189.8 Kg/mol and Ð of 2.4.

As mentioned above, the MMA polymerization proceeded in the presence of CTA-1 at three different AIBN ratios, keeping the MMA and CTA-1 ratios constant (Entries 1, 3, and 4). As it can be observed in Table 1 for these entries, the experimental molecular weight or *M_n_*_,SEC_ is at least twice higher than the theoretical values (*M_n_*_,Th_) and this discrepancy increases when the [AIBN] decreases. It is important to mention that the Đ values remained low (around 1.3). Based on these results, it can be assumed that the RAFT agent is not completely consumed during the polymerization using the studied conditions. The partial consumption of CTA-1 throughout the MMA polymerizations is evidenced in Figure 3. The PMMA of Entry 1 at 91% conversion was precipitated in distilled cold hexane to isolate the polymer, then the hexane was completely evaporated, and the residues were analyzed by ^1^H NMR. The spectrum of these residues is shown in Figure 3a, which shows the presence of the aromatic protons and a methine at a δ = 7.4, and 6.5 ppm, respectively. The spectrum lack of any signals corresponding to the backbone of PMMA. This result exhibits the presence of unreacted CTA-1 in the collected residues. On the other hand, the PMMA bearing the diphenylmethyl groups from the modified (or reacted according to the RAFT mechanism) CTA-1 was also analyzed by ^1^H NMR. Aromatic protons unequivocally derived from the diphenylmethyl groups bonded to the polymer chains can be clearly observed in Figure 3b. Other important signals, such as the -O-CH_3_ from the PMMA units (3.6 ppm) were directly assigned in the spectrum. Moreover, a signal at 6.45 ppm was observed, which could mean that part of the polymer chains are end-thiocarbonylthio terminated (PMMA-S-(C=S)-S-CH-(C_6_H_5_)_2_) and not all of them are middle-thiocarbonylthio functionalized, as stated by the RAFT mechanism for a symmetrical trithiocarbonate. A proposed reaction scheme of the above discussed is shown in Figure 4.

The NMR results from Figure 3 demonstrated the partial consumption of CTA-1, which explains the discrepancy between experimental and theoretical *M_n_*. It is worth mentioning that the RAFT polymerization is mainly featured by the total consumption of CTA during the process. However, the presence of unreacted CTA-1 during the RAFT polymerization of MMA is unusual and could be ascribed to the steric hindrance of the leaving diphenylmethyl group. We expect that the aromatic groups led to the formation of stable radicals which delay the β-scission during the addiction-fragmentation process, and only a part of CTA-1 acts for generating oligomeric chains with low monomer units. The presence of monomer units between thiocarbonylthio and leaving groups allows the β-scission therefore the control on the growth chains.

Based on the purpose of improving the *M_n_*_,SEC_/*M_n_*_,Th_ correlation, an experiment increasing the molar ratio [MMA]/[CTA-1] to 300:4 instead of 300:2 was performed to have a higher concentration of the controlling compound in the polymerization reaction, the AIBN proportion was maintained in 1. At this molar ratio (under same conditions: 60 °C, 8 h of reaction) the MMA conversion was 43% and unexpectedly the *M_n_*_,SEC_ (12.4 Kg/mol) still maintained a very significant difference with the theoretically calculated (around 3.2 Kg/mol), even this difference became bigger. Moreover, Ð value was not lowered, it was in the same range as all previous results of PMMA produced with CTA-1 (between 1.3 and 1.4). It is important to notice that in this experiment the *M_n_* calculated by SEC matches well with those calculated by ^1^H NMR (13.0 Kg/mol), indicating that most of the PMMA chains contain the CTA-1 aromatic functionality, even when the CTA-1 was not totally consumed during the reaction, as it was above explained. Integrated areas of phenyls groups signal around 7.4 ppm (assignation A in Figure 3b) and the methyl group of the PMMA signal at 3.6 ppm (assignation B in the same Figure) were used to calculate the *M_n_* by this equation *M_n_*_,NMR_ = [(Integral B/3)/(Integral A/20)] * 100 g/mol.

The unexpected lack of correlation between *M_n_*_,SEC_ and *M_n_*_,Th_ inspired the performance of three further experiments where the amount of CTA-1 was reduced (MMA and AIBN were kept constant), to study new ratios [CTA-1]/[AIBN] of 1:1, 0.5:1 and 0.25:1; giving unexpectedly better *M_n_*_,Exp_/*M_n_*_,Th_ correlations as the amount of CTA-1 was decreasing: 1.50, 1.37 and 1.16 values respectively. In Figure 5 these results are shown.

It was demonstrated in Figure 3 that CTA-1 was not completely consumed in the MMA polymerization using a ratio of [CTA-1]/[AIBN] = 2:1. It is well-known that molar ratios, where the RAFT agent is equal or lower than the initiator, are not common for RAFT polymerizations conditions, by which a small proportion of initiator is typically recommended to obtain low Ð values. However, the results obtained in this work indicate that using more quantity of AIBN in relation to the RAFT agent helps to higher consumption of the latter, and thus, the correlation *M_n_*_,Exp_/*M_n_*_,Th_ was improved and coming very closer to 1. The low Ð, around 1.3, were maintained in all five experiments reported in Figure 5.

Data from the polymerization of MMA, carried out at different concentration of CTA-1, clearly demonstrated the impact of [AIBN] in the correlation between the experimental and theoretical molecular weights of PMMAs. As observed, the SEC traces of PMMAs (Figure 5) shift towards higher molecular weights with the diminution of the [CTA]. It is well-known that the amount of AIBN can not only predict the number of dead chains into the RAFT polymerization but also the amount of active centers produced during the addition-fragmentation process [9,32]. Notwithstanding the above, the chemical structure and steric hindrance of CTA-1 could greatly diminish the transfer coefficient (*C_tr_*) of the chain agent and cause a slow re-initiation of leaving radicals R from CTA-1 giving rise to a retardation in the reversible deactivation process.

Besides the aromatic functionality of CTA-1 in the PMMA demonstrated by the ^1^H NMR analysis, a UV analysis was also carried out to corroborate that the trithiocarbonate (-S-(C=S)-S-) functionality of the CTA-1 was present in the polymer. The UV spectra in Figure 6a for CTA-1 and for a purified (precipitated 3 times in cold hexane) PMMA synthesized with CTA-1 show (both) a strong UV absorption corresponding to p/p * transition of the thiocarbonylthio moiety in the wavelength range 280–360 nm. Absorption of aromatic rings in the range of 215 to 275 nm is also observed in both spectra, suggesting that the diphenylmethyl group is bonded to the polymer. To further confirm the presence of the trithiocarbonate group in the middle of the PMMA chains, as expected for a RAFT polymerization made with a symmetrical trithiocarbonate [24], a PMMA synthesized with CTA-1 was treated with an excess of AIBN at 80 °C for 2.5 h to produce an exchange of polymeric chains by the group -C(CH_3_)_2_CN [33,34]. The resulting polymer was analyzed by SEC and the results showed the expected shift towards lower molecular weights region of the chromatogram (Figure 6b). The radical attack induced a decrease of *M_n_* from 29.2 to 18.5 Kg/mol, indicating the cleavage of the polymer in the trithiocarbonate moiety.

### 2.3. Glycidyl Methacrylate and Ethyl Methacrylate

Both CTA-1 and CTA-2 were also utilized in the polymerization of derivative methacrylates (GMA and EMA). The reactions were performed under similar conditions than polymerization of MMA and the ratio [Monomer]/[CTA]/[AIBN] was maintained constant (300:2:1). Data from SEC analysis (Entry 5) from the polymerization of GMA with CTA-1 showed a *M_n_*_,SEC_ = 45.1 Kg/mol and Ð = 1.9, while the theoretical *M_n_*_,Th_ was calculated to 14.6 Kg/mol. Similar to the case of MMA, this *M_n_*_,SEC_/*M_n_*_,Th_ correlation was not close to 1 value, and the resulting Ð cannot be considered as a good value for RAFT. Nonetheless, to the best of our knowledge, this is the first example reported for a homogeneous (bulk or solution) GMA polymerization kind of controlled by a symmetrical trithiocarbonate, in absence of an auto-acceleration effect. The term “controlled” was daring to be used in this case since in the polymerization of GMA with the reference CTA-2 (Entry 6), the resultant PGMA was insoluble in THF, chloroform, acetone, and toluene. We presume the formation of gel due to crosslinking and therefore the sample is neither analyzed by SEC nor NMR. Crosslinked GMA is typically obtained through conventional radical polymerization in bulk conditions [35], and thus, highlights the relevance of being able to synthesize PGMA in a rather controlled way by employing a symmetrical trithiocarbonate, in this case, CTA-1.

For the polymerization of EMA with CTA-1 and CTA-2, data from SEC analyses were displayed in Entries 7, and 8, respectively (See Table 1). Similar to the polymerization of MMA with CTA-1, results from SEC analysis of PEMA obtained with CTA-1 showed a *M_n_*_,SEC_ = 30.5 Kg/mol, which is approximately twice higher than the theoretical (*M_n_*_,Th_ = 13.5 Kg/mol) and a low Ð = 1.3 with a good conversion ca. to 80%. The low Ð value indicates a certain control exercised by the CTA-1 on the polymerization of EMA. In the case of the polymerization of EMA with the reference CTA-2, data from SEC analysis exhibited a *M_n_*_,SEC_ ca. to 200 Kg/mol and broad molecular weight distribution when compared with the resulting parameters using CTA-1. These last results indicate the expected poor (or non-existent) control of the polymerization of EMA using the reference CTA-2.

### 2.4. Styrene and Butyl Acrylate

Additionally, CTA-1 and CTA-2 were also utilized in the polymerization of St and BuA. Reactions were performed under similar conditions than PMMA (T = 60 °C, t = 15 h) and the ratio [Monomer]/[CTA]/[AIBN] = 300:2:1 was maintained constant to compare the results from each experiment. Data are summarized in Table 2.

As observed in Entries 9 and 11, the polymerization of both monomers (St and BuA) in the presence of CTA-1 exhibited an extended inhibition period featured by a conversion of 3% after 15 h of reaction and *M_n_*_,SEC_ values lower than 0.5 Kg/mol, see Figure 7 for St case. As abovementioned, the great stability of the leaving group (diphenylmethyl radical) does not re-initiate the propagation step due besides to the steric effect. For this reason, high inhibition was observed and only oligomers were acquired. The monomers St and BuA were polymerized in the presence of reference CTA-2 prepared in our lab, see Entries 10 and 12 in Table 2. Data from SEC analyses of both experiments revealed a good control of CTA-2 in the polymerization of St (*M_n_*_,SEC_ = 3.6 Kg/mol, Đ = 1.3) and BuA (*M_n_*_,SEC_ = 21.4 Kg/mol, Đ = 1.2). These results matched with reported data from the polymerization of St and BuA in the presence of the well-studied CTA-2 [31,36,37,38,39].

### 2.5. Block Copolymerizations

The RAFT technique is featured by the capacity of previously synthesized macro-CTA to be chain extended for obtaining copolymers with defined structure and special properties. For highlighting this characteristic, the PMMA from the Entry 4 (*M_n_*_,SEC_ = 21.7 Kg/mol, Đ = 1.3) was utilized as macro-CTA for preparing copolymers by addition of St and BuA. Considering that thiocarbonylthio functionality is mainly localized in the middle of the macro-CTA chains, triblock copolymers (ABA) PMMA-*b*-PSt-*b*-PMMA and PMMA-*b*-PBuA-*b*-PMMA were expected. Table 3 summarizes the results related to the synthesis of the block copolymers using different reaction conditions.

The block copolymerizations using the CTA-1 functionalized PMMA from Entry 4 as macro-CTA were performed at a molar ratio of [Monomer]/[macro-CTA]/[AIBN] = 608:1:1.3 in the case of St and 509:1:1.3 in the case of BuA (or a mass ratio = 300:100:1 in both cases), at a temperature of 60 °C for 15 h. The consumption of St (65%) and BuA (96%) was calculated by ^1^H NMR. For the case of St as the monomer, data from SEC analyses of the resulting copolymer demonstrated that the *M_n_*_,SEC_ = 66.9 Kg/mol is close to the *M_n_*_,Th_ = 63.9 Kg/mol with a Đ = 1.2. The composition of this triblock copolymer was calculated by ^1^H NMR resulting in 69.5 wt.% of PSt and 30.5 wt.% of PMMA, which agrees with the expected composition according to the recipe and the monomer consumption. On the other hand, in Entry 14 BuA was used to synthesize a triblock copolymer PMMA-*b*-PBuA-*b*-PMMA with *M_n_*_,SEC_ = 76.4 Kg/mol and a Đ = 1.4. In this experiment, the value for *M_n_* experimental is also closed to the theoretical one. The composition of the resultant triblock copolymer was 76 wt.% of PBuA and 24 wt.% of PMMA as calculated by ^1^H NMR.

The SEC traces of both synthesized copolymers exhibited in Figure 8 showed a monomodal distribution with a shift towards a higher molecular weight region, which corroborate the insertion of novel blocks to the PMMA “first” block and the formation of the expecting triblock copolymers. In the case of PMMA-*b*-PSt-*b*-PMMA, the SEC traces from the refractive index and UV detectors are overlapped indicating the presence of PSt in all the detected chains (Figure 8a). For the PMMA-*b*-PBuA-*b*-PMMA, only the trace from the refractive index detector was observed because PBuA is undetected under this condition (Figure 8b). These findings represent evidence of the “living” characteristic of PMMA obtained by the RAFT conditions polymerization of this methacrylic monomer in the presence of the novel CTA-1.

## 3. Materials and Methods

Reagents: the monomers methyl methacrylate (MMA, 99%), ethyl methacrylate (EMA, 99%), glycidyl methacrylate (GMA, 97%), styrene (St, ≥99%), and butyl acrylate (BuA, ≥99%) were distilled under vacuum before use. Carbon disulfide (99.9%), resin amberlyst A-26(OH), benzyl bromide (98%), bromodiphenylmethane (95%), and anhydrous magnesium sulfate (≥99.5%) were used as received. AIBN (98%) was recrystallized twice from ethanol before use. CDCl_3_ and THF (HPLC grade) were also used as received. All reagents were purchased from Sigma-Aldrich (Toluca, Mexico).

Characterization: Size exclusion chromatography (SEC). The molecular weight characteristics of polymers were determined by SEC using a Hewlett-Packard instrument (HPLC series 1100) equipped with UV light and refractive index detectors. A PLGel mixed column was used. Calibration was carried out with polystyrene and poly(methyl methacrylate) standards and THF (HPLC grade) was used as eluent at a flow rate of 1 mL/min.

Nuclear magnetic resonance (NMR). The chemical structure of CTA, polymers, and the conversion rate of the monomers was tracked via ^1^H NMR using a Bruker Avance III HD 400N spectrometer (with a 5 mm multinuclear BB-decoupling probe, direct detection with Z grad). The analyses were performed at 25 °C and the samples were diluted in CDCl_3_.

Ultraviolet spectroscopy (UV) was carried out in a Varian Cary 100 UV/Vis spectrophotometer (Agilent Technologies) to corroborate the presence of thiocarbonylthio groups [-S-(C=S)-S-] within the polymer chains of functionalized RAFT polymers.

Synthesis of di(diphenylmethyl) trithiocarbonate (CTA-1): 15 g of amberlyst A-26 (OH) previously dried at 110 °C were collocated into a three-neck round flask equipped with a magnetic stirring, condenser, and addition funnel under argon flow. 50.40 g (0.663 mol) of carbon disulfide were added, and the resulting suspension was stirred at room temperature for 10 min. The color of the resin changed from yellowish to red indicating the formation of thiocarbonylthio group. After that, 3.76 g (0.015 mol) of bromodiphenylmethane were added and the reaction mixture was stirred under reflux for 10 h in an inert atmosphere. Then, the mixture was filtered and washed three times with THF. The filtrate was dried over anhydrous magnesium sulfate and the solvent evaporated under reduced pressure to obtain the crude product. The CTA-1 was acquired by recrystallization in hexane. Yield of 92% (3.05 g, 6.9 * 10^−3^ mol) after purification was obtained. ^1^H NMR. δ: 6.5 (s, 2H, S-CH-(C_6_H_5_)_2_), 7.2-7.4 (m, 20H, ArH). ^13^C NMR. δ: 59 (S-CH-(C_6_H_5_)_2_), 127-139 (ArC), 219 (C=S).

The synthesis of di(benzyl) trithiocarbonate (CTA-2): 15 g of amberlyst A-26(OH) before dried at 110 °C were added into a three-neck round flask equipped with a magnetic stirring, addition funnel, and condenser under argon flow. 56.70 g (0.746 mol) of carbon disulfide were added and the suspension was stirred at room temperature for 10 min. The color of the resin turns from yellowish to red, pointing out the formation of thiocarbonylthio group. After that, 5.11 g (0.030 mol) of benzyl bromide were added and the reaction mixture was stirred under reflux for 10 h. Then, the mixture was filtered and washed three times with THF. The filtrate was dried over anhydrous magnesium sulfate and the solvent evaporated under reduced pressure to afford the crude product. The trithiocarbonate was purified by column chromatography using a mixture of petroleum ether:benzene (9:1) as eluent. Yield of 21% (0.913 g, 3.15 * 10^−3^ mol) after purification was obtained. ^1^H NMR. δ: 4.6 (s, 4H, S-CH_2_-C_6_H_5_), 7.2–7.4 (m, 10H, ArH). ^13^C NMR. δ: 41 (S-CH_2_-C_6_H_5_), 127–135 (ArC), 223 (C=S).

Hompolymerization reactions: in a typical RAFT polymerization reaction, Entry 1 in Table 1, a stock solution of MMA (9.0 g, 0.09 mol), CTA-1 (0.265 g, 5.99 * 10^−4^ mol) and AIBN (0.049 g, 2.99 * 10^−4^ mol) (molar ratio of 300:2:1) was prepared. Aliquots of 1.5 g approximately were transferred to six ignition tubes, degassed with three freeze-evacuate-thaw cycles, and sealed with flame under vacuum. The tubes were then heated to 60 °C in an oil bath and left for different reaction times (15 h was the final one). An aliquot of each reaction crude was analyzed by ^1^H NMR to calculate monomer conversion. The isolated polymer was analyzed by SEC to estimate the number-average molecular weight (*M_n_*) and molecular weight dispersity (Ð) of the resulting polymer.

Based on similar conditions, other MMA experiments were performed varying the [CTA-1]/[AIBN] ratio (Entries 3 and 4 in Table 1). Moreover, polymerizations of derivative methacrylate monomers such as EMA and GMA were performed using CTA-1 under similar reaction conditions. Further, all monomers were tested using CTA-2 as a reference controller.

Block copolymerization reactions: in a typical reaction, Entry 13 in Table 3, a solution of functionalized PMMA as macro-CTA (0.50 g, 2.30 * 10^−5^ mol), styrene (1.50 g, 0.014 mol), and AIBN (5 mg, 3.05 * 10^−5^ mol) was prepared and transferred to a tube, which was degassed with three freeze-evacuate-thaw cycles and sealed with flame under vacuum. The tube was heated to 60 °C in an oil bath and left for 15 h. Reaction crude was analyzed by ^1^H NMR to calculate monomer conversion. The resulting copolymer was isolated by precipitation in cold hexane and vacuum-dried, after that, it was analyzed by SEC to estimate macromolecular parameters.

## 4. Conclusions

The di(diphenylmethyl) trithiocarbonate or CTA-1 was successfully synthesized in a yield of 92%, purified and characterized by ^1^H and ^13^C NMR. This symmetrical trithiocarbonate was used for the first time as chain transfer agent under RAFT polymerization conditions for methacrylic monomers, as well as styrene and butyl acrylate. Methyl methacrylate polymerization in presence of CTA-1 showed low molecular weight dispersity values, below 1.5 and a “living” behavior. Contrary to previously reported for RAFT, atypically high [AIBN]/[CTA-1] molar ratios were required for accessing predictable molecular weights. We also provided important insights towards the methacrylates controlled polymerizations in homogeneous media using a symmetrical trithiocarbonate as RAFT agent. It was proved that CTA-1 inhibited the polymerization of styrene and butyl acrylate because of the great stability and steric effect of the leaving group (diphenylmethyl radical), which did not re-initiate the propagation step in the reaction. Poly(methy methacrylate)s, prepared in presence of CTA-1, were successful macro-CTAs to synthesize well-defined block copolymers by sequential polymerization using styrene and butyl acrylate as co-monomers.

## Figures and Tables

**Figure 1 molecules-26-04618-f001:**
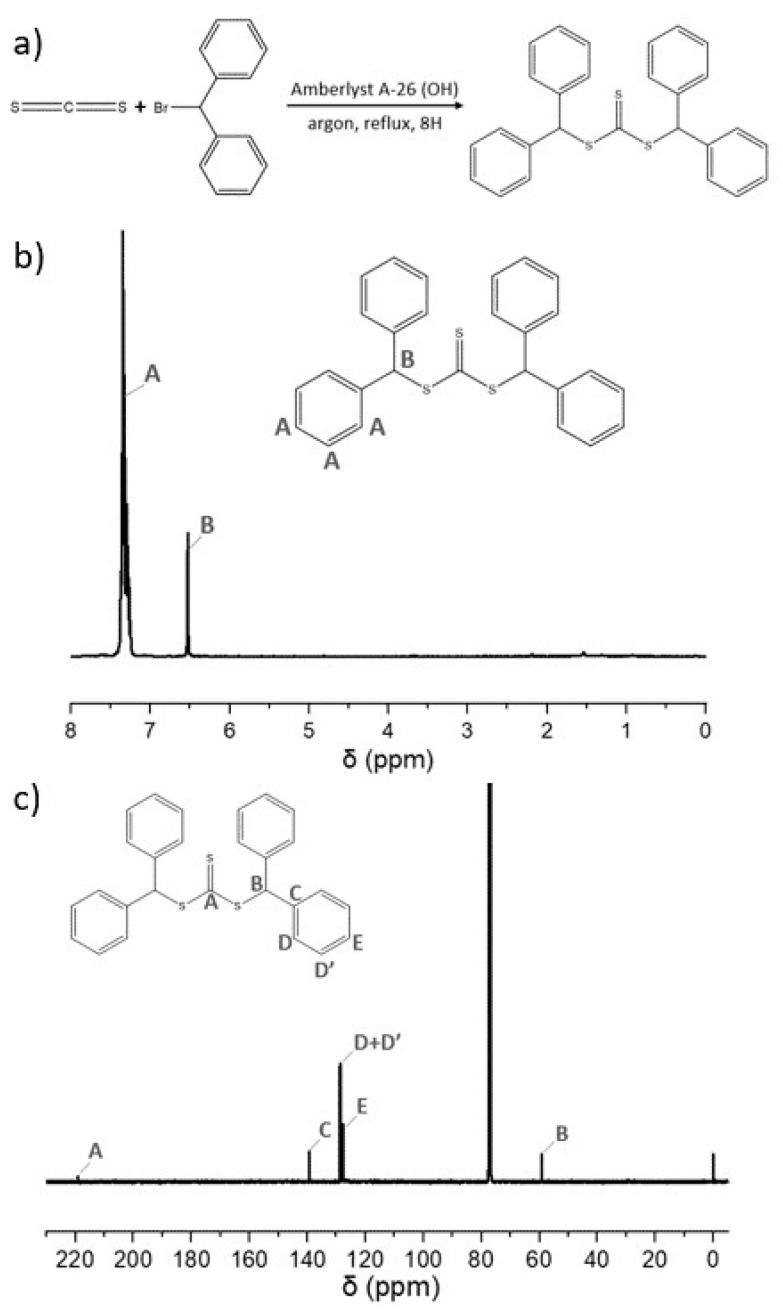
(**a**) Equation reaction to synthesize the di(diphenylmethyl) trithiocarbonate (CTA-1), (**b**) its corresponding ^1^H NMR and (**c**) ^13^C NMR spectra.

**Figure 2 molecules-26-04618-f002:**
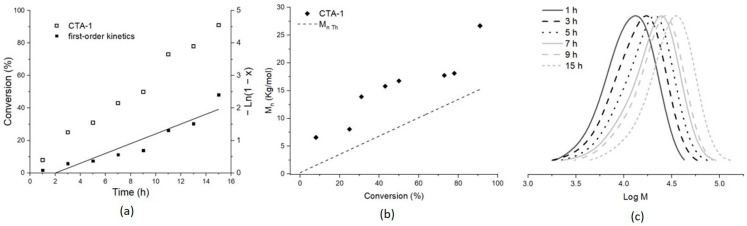
MMA polymerization using CTA-1: (**a**) Rate of conversion (inlet: first-order kinetics), (**b**) *M_n_*_,SEC_ as a function of conversion, and (**c**) SEC traces of PMMA obtained at a ratio [MMA]/[CTA-1]/[AIBN] = 300:2:1.

**Figure 3 molecules-26-04618-f003:**
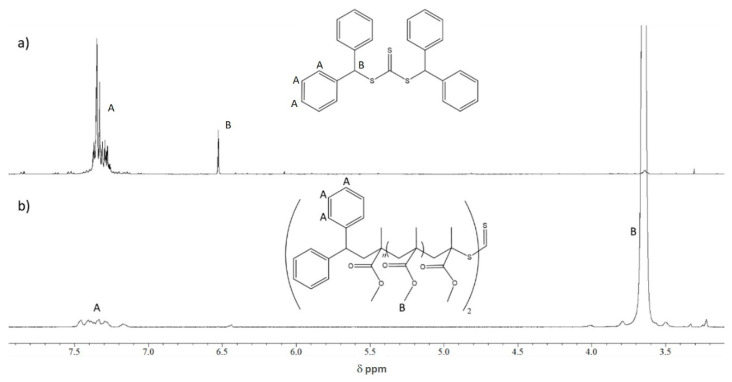
(**a**) ^1^H NMR spectrum of residual CTA-1 recovery after purification of PMMA from Entry 1; (**b**) ^1^H NMR spectrum of isolated PMMA from Entry 1 demonstrating its functionalization with aromatics groups.

**Figure 4 molecules-26-04618-f004:**
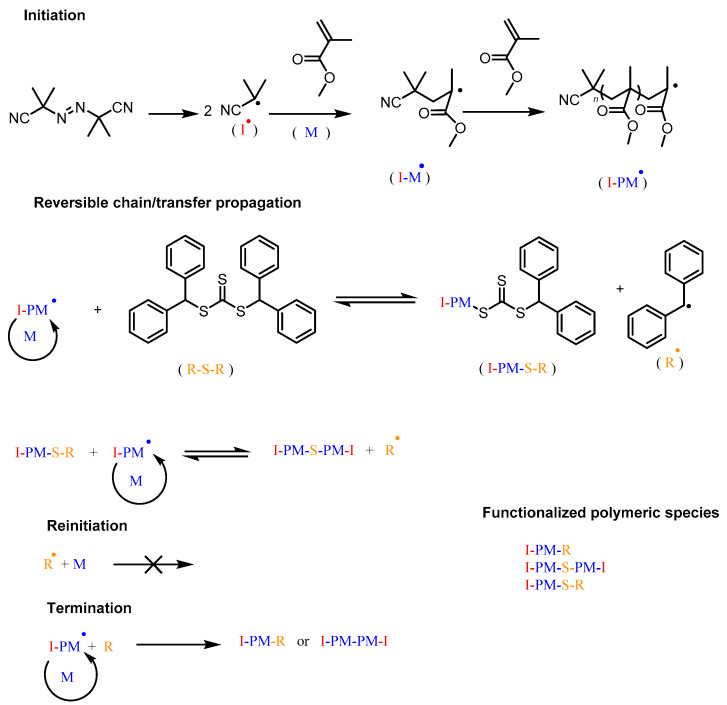
Plausible mechanism reaction for bulk MMA polymerization in presence of CTA-1 as controller and AIBN as initiator.

**Figure 5 molecules-26-04618-f005:**
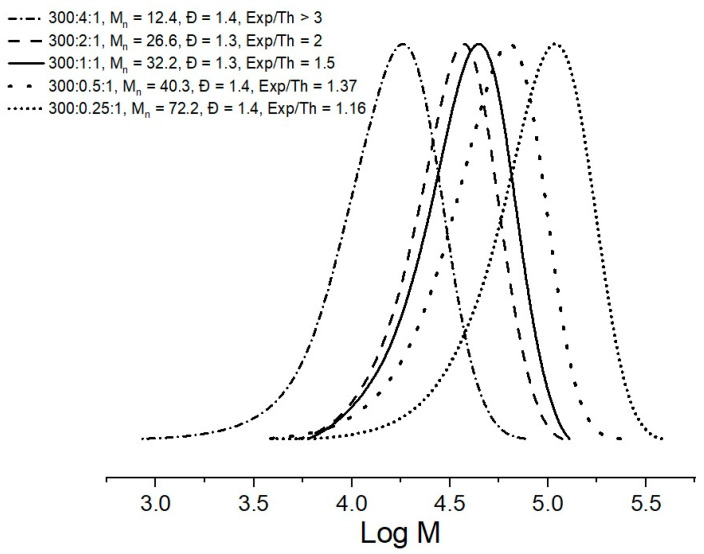
SEC traces for MMA polymerizations in presence of CTA-1 at different [CTA]/[AIBN] molar ratios.

**Figure 6 molecules-26-04618-f006:**
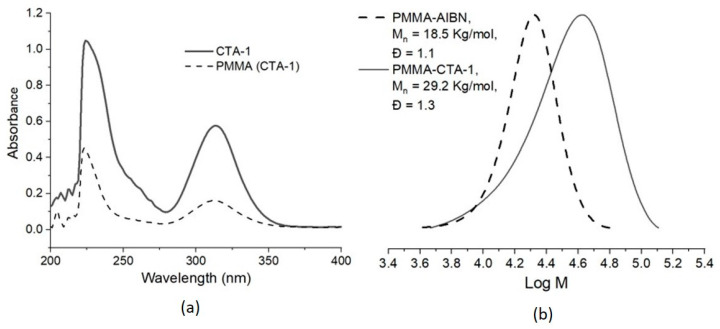
(**a**) UV analysis, and (**b**) AIBN cleavage of a PMMA synthesized in presence of CTA-1.

**Figure 7 molecules-26-04618-f007:**
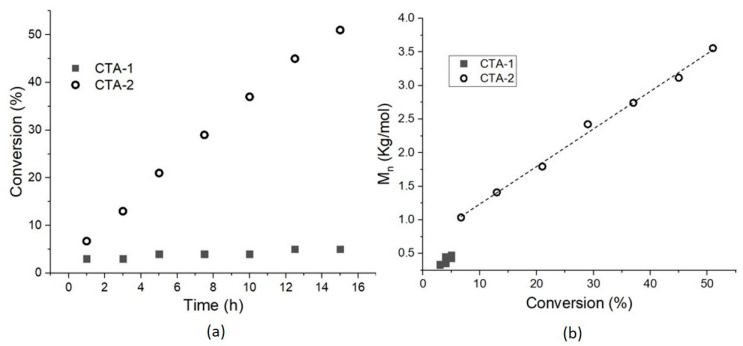
Polymerization of St using CTA-1 and CTA-2; (**a**) rate of conversion and (**b**) *M_n_* as a function of conversion.

**Figure 8 molecules-26-04618-f008:**
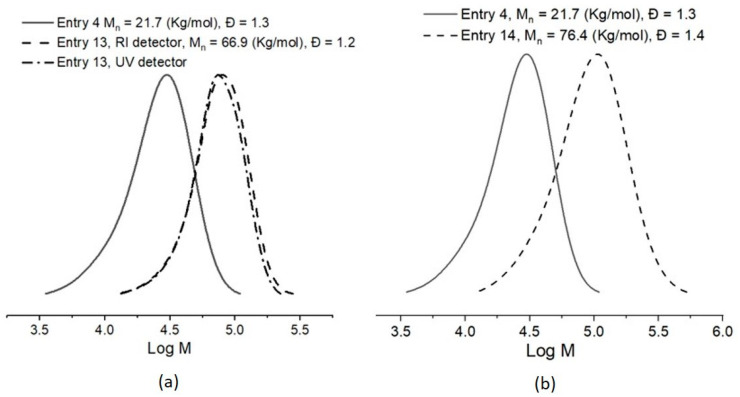
SEC traces for block copolymerizations; (**a**) entry 13, PMMA-*b*-PSt-*b*-PMMA; (**b**) entry 14, PMMA-*b*-PBuA-*b*-PMMA.

**Table 1 molecules-26-04618-t001:** Reaction conditions and results of methacrylic monomers polymerizations.

Entry	CTA	Monomer	Molar Ratio [Monomer]/[CTA]/ [AIBN]	^A^ Conversion (%)	^B^*M_n_*_,Th_ (Kg/mol)	^C^ *M_n_*_,SEC_ (Kg/mol)	Ð
1	CTA-1	MMA	300:2:1	91	13.9	26.6	1.3
2	CTA-2	MMA	300:2:1	82	12.4	189.7	2.4
3	CTA-1	MMA	300:2:0.66	70	10.7	25.8	1.3
4	CTA-1	MMA	300:2:0.33	54	8.3	21.7	1.3
5	CTA-1	GMA	300:2:1	96	14.6	45.1	1.9
6	CTA-2	GMA	300:2:1	>95	^D^ n.d.	^D^ n.d.	^D^ n.d.
7	CTA-1	EMA	300:2:1	79	13.5	30.5	1.3
8	CTA-2	EMA	300:2:1	>95	15.4	189.3	1.9

T = 60 °C, t = 15 h, ^A^ determined by ^1^H NMR, ^B^ calculated by *M_n_*_,Th_ = ((mass of monomer) * (fractional conversion)/(moles of RAFT agent)) + *M*_chain ends_ [*M*_chain ends_ = fragment AIBN + fragment CTA-1 = 235 g/mol], ^C^ determined by SEC with PMMA calibration, ^D^ not determined due to poor solubility.

**Table 2 molecules-26-04618-t002:** Reaction conditions and result of styrene and butyl acrylate polymerizations.

Entry	CTA	Monomer	^A^ Conversion (%)	^B^ *M_n_*_,Th_ (Kg/mol)	^C^ *M_n_*_,SEC_ (Kg/mol)	Ð
9	CTA-1	St	3	0.7	0.4	1.1
10	CTA-2	St	51	8.1	3.6	1.3
11	CTA-1	BuA	3	0.7	0.3	1.1
12	CTA-2	BuA	95	18.4	21.3	1.2

Molar ratio [Monomer]/[CTA]/[AIBN] = 300:2:1, T = 60 °C, t = 15 h, ^A^ determined by ^1^H NMR, ^B^ calculated by *M_n_*_,Th_ = ((mass of monomer) * (fractional conversion)/(moles of RAFT agent)) + *M*_chain ends_ [*M*_chain ends_ = fragment AIBN + fragment CTA-1 = 235 g/mol], ^C^ determined by SEC with PSt calibration.

**Table 3 molecules-26-04618-t003:** Reaction conditions and results of the chain extension and block copolymerizations.

Entry	Macro-CTA	*M_n_*_,SEC_ (Kg/mol), Ð	Monomer	T (°C), t (h)	Molar Ratio [Monomer]/ [macro-CTA]/ [AIBN]	^A^ Conversion (%)	^B^ *M_n_*_,Th_ (Kg/mol)	^C^ *M_n_*_,SEC_ (Kg/mol)	Ð
13	Entry 4	21.7, 1.3	St	60, 15	608:1:1.3	65	63.9	66.9	1.2
14	Entry 4	21.7, 1.3	BuA	60, 15	509:1:1.3	96	84.1	76.4	1.4

^A^ Determined by ^1^H NMR, ^B^ calculated by *M_n_*_,Th_ = ((mass of monomer) * (fractional conversion)/(moles of macro-CTA)) + *M*_macro-CTA_, ^C^ PMMA calibration.

## Data Availability

The data that support the findings of this study are available within the article.

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
