# Peer review of "Controlled (Co)Polymerization of Methacrylates Using a Novel Symmetrical Trithiocarbonate RAFT Agent Bearing Diphenylmethyl Groups"

_molecules, 2021, doi:10.3390/molecules26154618_

Round 1

Reviewer 1 Report

In this manuscript, the authors describe the synthesis of a novel symmetric trithiocarbonate bearing twin diphenylmethyl R-groups, which was utilized for the RAFT polymerization of methyl methacrylate bearing an internal trithiocarbonate. While decently narrow molecular weight distributions were obtained for all systems, the level of molecular weight control was found to be sensitive to the relative concentration of [AIBN]/[CTA], with increasing ratios corresponding to increasing control. This finding was hypothesized to arise from incomplete CTA consumption at low [AIBN], evidenced by the presence of unreacted CTA after polymerization. Otherwise, the system presented herein features all the expected characteristics of a successful RAFT polymerization.

I believe the manuscript presents work of an acceptable level of detail and novelty for publication in Molecules, and it will be suitable for publication after a couple supplementary experiments are performed. Methacrylates are an important monomer class, and the availability of a symmetric CTA could certainly prove useful. The ease of synthesizing the novel CTA is also a very nice feature. Of course, the unusual results regarding molecular weight control are the biggest question mark over this system. Considering this manuscript primarily reports new methodology, I appreciate that the authors performed initial experiments to interrogate its mechanistic features. If the mechanism is the subject of ongoing or future studies, I would suggest using NMR to study the kinetics of R-group consumption since it features a distinct chemical shift, as well as using MALDI to see if the end-groups change as [AIBN] changes. Nevertheless, my comments and suggestions with regards to the present work are as follows:

P.2, L.61: Dithiobenzoates are the most reactive of the commonly used CTAs (see Macromolecules 2012, 45, 5321–5342). The authors should consider revising the statement to avoid suggesting that trithiocarbonates are more reactive.

P.2, L.72: The introduction here suggests that the poor performance of symmetric CTAs is a consequence of the symmetric structure, rather than the R-group. However, the cited examples of symmetric CTAs all feature unsuitable R-groups for methacrylate polymerization, which I believe is more consequential than the symmetrical structure. The authors should briefly address this point and, if the symmetry of the CTA is also believed to be important, provide a more detailed discussion of such.

P.5, Figure 2: Is the dashed line a simple linear fit of the data points? If so, it should be removed and replaced with a line depicting Mn,Th as a function of conversion.

P.6: Was the amount of residual CTA relative to the feed quantity determined and if so, does it fully explain the discrepancy between theoretical and experimental molecular weight? If not, this data should be obtained and included as a point of discussion.

P.8: The increase in MW control with additional AIBN is an intriguing finding and deserves further consideration. While I believe a detailed mechanistic study is beyond the scope of this manuscript, it would be nice to see the results of a kinetics experiment using [MMA]/[CTA]/[AIBN] =300/0.25/1.00 in order to better compare this system to the results shown in Figure 2.

Minor comments

P.7, Figure 4: There is a missing methyl group in the monomer structure (i.e., the depicted structure is methyl acrylate, not MMA).

Author Response

English language and style

English language and style are fine/minor spell check required.

Answer: the manuscript was thoroughly revised in terms of English language, style and grammar mistakes, ensuring also that the appropriate abbreviations are used. The mistakes we found after this revision were corrected and indicated in the latest version of manuscript for revision.

Comments and suggestions

In this manuscript, the authors describe the synthesis of a novel symmetric trithiocarbonate bearing twin diphenylmethyl R-groups, which was utilized for the RAFT polymerization of methyl methacrylate bearing an internal trithiocarbonate. While decently narrow molecular weight distributions were obtained for all systems, the level of molecular weight control was found to be sensitive to the relative concentration of [AIBN]/[CTA], with increasing ratios corresponding to increasing control. This finding was hypothesized to arise from incomplete CTA consumption at low [AIBN], evidenced by the presence of unreacted CTA after polymerization. Otherwise, the system presented herein features all the expected characteristics of a successful RAFT polymerization.

I believe the manuscript presents work of an acceptable level of detail and novelty for publication in Molecules, and it will be suitable for publication after a couple supplementary experiments are performed. Methacrylates are an important monomer class, and the availability of a symmetric CTA could certainly prove useful. The ease of synthesizing the novel CTA is also a very nice feature. Of course, the unusual results regarding molecular weight control are the biggest question mark over this system. Considering this manuscript primarily reports new methodology, I appreciate that the authors performed initial experiments to interrogate its mechanistic features. If the mechanism is the subject of ongoing or future studies, I would suggest using NMR to study the kinetics of R-group consumption since it features a distinct chemical shift, as well as using MALDI to see if the end-groups change as [AIBN] changes. Nevertheless, my comments and suggestions with regards to the present work are as follows:

1) P.2, L.61: Dithiobenzoates are the most reactive of the commonly used CTAs (see Macromolecules 201245, 5321–5342). The authors should consider revising the statement to avoid suggesting that trithiocarbonates are more reactive.

Answer: We totally agree with the reviewer. The dithioesteres, specially the dithiobenzoates, are the most effective RAFT agents due to their high transfer constants (Ctr) [Polymer 2008, 49, 1079]. In fact, the dibenzyl trithiocarbonate (or CTA-2 in this manuscript) presented a Ctr = 53 for the styrene polymerization at 80°C [Polym. Sci. Ser. A. 2007, 49, 108], while a Ctr > 500 has been reported for a dithiobenzoate in the styrene polymerization at 60 °C [Macromolecules 2003, 36, 2256].

In that sentence in specific we wanted to say that trithiocarbonates are among the most reactive RAFT agents, but excluding the dithiobenzoates.

In the new version of the manuscript this fact was clarified.

2) P.2, L.72: The introduction here suggests that the poor performance of symmetric CTAs is a consequence of the symmetric structure, rather than the R-group. However, the cited examples of symmetric CTAs all feature unsuitable R-groups for methacrylate polymerization, which I believe is more consequential than the symmetrical structure. The authors should briefly address this point and, if the symmetry of the CTA is also believed to be important, provide a more detailed discussion of such.

Answer: we partially agree with the reviewer; we believe that if the control of methacrylic monomers depended 100% on the nature of the R group, then there would be reports of symmetric trithiocarbonates with those good R groups (-CH(Ph)-CN for example) perfectly controlling the polymerization of MMA, and there are none. However, we agree with the reviewer that the role that play the R group is more important than what is read in that part of the manuscript.

A combined effect of both factors is what we believe happens in this situation. A good experiment that we will consider from this comment is the synthesis of CN-(Ph)CH-S-(C=S)-S-CH(Ph)-CN as RAFT agent and its evaluation in MMA polymerizations. But this new molecule is out of the purpose of this manuscript.

In the new version of the manuscript a brief text of the point here discussed was added. 

3) P.5, Figure 2: Is the dashed line a simple linear fit of the data points? If so, it should be removed and replaced with a line depicting Mn,Th as a function of conversion.

Answer: the dashed line was replaced as properly suggested.

4) P.6: Was the amount of residual CTA relative to the feed quantity determined and if so, does it fully explain the discrepancy between theoretical and experimental molecular weight? If not, this data should be obtained and included as a point of discussion.

Answer: according to the NMR analysis presented in the Figure 3, and also based on the yellowing of the resulting polymer, it is evident that part of the CTA-1 is incorporated into the polymer chains and the other part does not react. However, due to the very small amount (g) of the RAFT agent used in our experiments, we did not believe it was reliable to make the calculation indicated by the reviewer. In this manuscript, unfortunately we cannot determine that relationship between the CTA-1 consumed (or the one that does not react) with the Mn of the polymer. This excellent comment will be taken into account for our next publication on this topic in which we will design an experiment that involves a high amount of CTA-1 (g) and that allows us to quantify it at the end of the precipitation and that the determined value does not fall within of a possible experimental error.

5) P.8: The increase in MW control with additional AIBN is an intriguing finding and deserves further consideration. While I believe a detailed mechanistic study is beyond the scope of this manuscript, it would be nice to see the results of a kinetics experiment using [MMA]/[CTA]/[AIBN] =300/0.25/1.00 in order to better compare this system to the results shown in Figure 2.

Answer: we completely agree with the reviewer that other kinetic studies at the different [MMA]/[CTA-1]/[AIBN] molar ratios would be useful to reinforce this manuscript. We are even aware that kinetic studies, as well as evaluate different [CTA-1]/[AIBN] molar ratios, of the other monomers reported would also support a broader discussion of the manuscript. However, we submitted this first article related to CTA-1 with the purpose of making it known and testing the viability that methacrylate-based block copolymers can be prepared from this molecule. Fully dedicated kinetic studies on MMA (and other monomers such as DMAEMA or 2,2,2-trifluoroethyl methacrylate) are not the scope of this manuscript, nevertheless we are still working in the laboratory on this topic and much more results will soon be submitted, with other type of characterization such as MALDI-TOF and DOSY.

6) P.7, Figure 4: There is a missing methyl group in the monomer structure (i.e., the depicted structure is methyl acrylate, not MMA).

Answer: the reviewer is totally correct in this comment; we apologize for this structure mistake. In the new version of the manuscript we modified the Figure 4 in order to correct this and other related mistakes.

The reviewer is thanked for its valuable comments and corrections.

We believe important to mention that NMR experiments such as the reaction of AIBN + CTA-1 or AIBN + CTA-1 + MMA (equimolar) carried out directly in the NMR equipment, or MALDI-TOF analysis, are in process to be thoroughly revised for future publications. An asymmetric trithiocarbonate containing a diphenylmethyl R group is also planned to be soon published as RAFT agent for “more activated monomers” polymerizations.

Reviewer 2 Report

The paper: “Controlled (co)polymerization of methacrylates using a novel symmetrical trithiocarbonate RAFT agent bearing diphenyl-3 methyl groups” describes the synthesis of novel symmetrical RAFT agent and its utilization for polymerization of various monomers. I need to admit that I really like this research. I appreciate the efforts undertaken to explain the mechanism of the polymerization. However, the paper could be improved in some aspects.  

  1. I cannot agree with the statements regarding the activity of the monomers (line 56-60). It is suggested that monomers with conjugated double bonds are very active. In my opinion this is too general statement, as active centers in case of polymerization of some monomers with conjugated double bonds are stabilized by resonance effect (e.g. styrene, butadiene etc.).
  2. It would be beneficial to somehow estimate the total amount/concentration of radicals produced from AIBN for the tests performed for different molar ratios of AIBN/CTA. It seems that this plays a crucial role and may explain the differences between theoretical and experimental Mn. The total concentration of active centers produced from primary radicals may equals CTA concentration e.g. in case of polymerization conduced at following ratio of reagents 300:0.25:1. I wonder if most of R groups could be replaced (at this condition) by the fragments coming from chains started by primary radicals? Furthermore, in classical RAFT the total number of radicals produced from initiators equals the total number of dead chains (irreversibly terminated). May the authors comment on that?
  3. In my opinion the linear fit in Fig. 2A is misleading. After around 10 h the polymerization seems to accelerate as there is more of active centers. In case of Fig. 2 B most of point could be fitted rather by logarithmic fit, which is typical for polymerizations with slow initiation.
  4. I have some comments to Fig. 4. In case of step describing “RAFT equilibrium” the authors wrote in the bracket: “is this stage reversible?” I would like to ask: who should answer this question? Reviewer, readers, or authors? The scheme suggests that reinitiation cannot by realized by leaving R groups, while in the main text it is suggested that some “part of CTA-1 acts for generating oligomeric chains with low monomer units” (line 199 -200). If really there is no reinitiation then we cannot say that this is RAFT mechanism. It is also interesting what is happening with the leaving R radicals. Is it possible recombination of two R radicals?
  5. Mass ratios of monomer, macro-CTA and AIBN in Table 3 are not informative. I suggest presenting molar ratios.
  6. It is quite surprising that polymerization of butyl acrylate was ineffective as the propagation rate constant in case of this monomer is rather high. What could be the explanation of this phenomenon?  

Author Response

Moderate English changes required.

Answer: the manuscript was thoroughly revised in terms of English language, style and grammar mistakes, ensuring also that the appropriate abbreviations are used. The mistakes we found after this revision were corrected and indicated in the latest version of manuscript for revision.

Comments and suggestions

The paper: “Controlled (co)polymerization of methacrylates using a novel symmetrical trithiocarbonate RAFT agent bearing diphenyl-3 methyl groups” describes the synthesis of novel symmetrical RAFT agent and its utilization for polymerization of various monomers. I need to admit that I really like this research. I appreciate the efforts undertaken to explain the mechanism of the polymerization. However, the paper could be improved in some aspects.  

1) I cannot agree with the statements regarding the activity of the monomers (line 56-60). It is suggested that monomers with conjugated double bonds are very active. In my opinion this is too general statement, as active centers in case of polymerization of some monomers with conjugated double bonds are stabilized by resonance effect (e.g. styrene, butadiene etc.).

Answer: in some reviews of the RAFT process give this classification of more activated monomers (MAM) vs less activated monomers (LAM), with a similar definition to the one we gave in the manuscript, but in this case we think that the reviewer is correct in his comment and that this information was used in a very general way in our manuscript; so all this information related to MAM´s and LAM´s was corrected in the manuscript in order to avoid generality and be more specific.  

2) It would be beneficial to somehow estimate the total amount/concentration of radicals produced from AIBN for the tests performed for different molar ratios of AIBN/CTA. It seems that this plays a crucial role and may explain the differences between theoretical and experimental Mn. The total concentration of active centers produced from primary radicals may equals CTA concentration e.g. in case of polymerization conduced at following ratio of reagents 300:0.25:1. I wonder if most of R groups could be replaced (at this condition) by the fragments coming from chains started by primary radicals? Furthermore, in classical RAFT the total number of radicals produced from initiators equals the total number of dead chains (irreversibly terminated). May the authors comment on that?

Answer: we agree with the reviewer for his kind comment. As mentioned by Favier et al. Macrom. Rapid Commun., 2006, 27, 653-692 and Perrier, Macromolecules, 2017, 50, 7433-7447, the number of radicals from the decomposition of initiator determines the amount of dead chains in the system. Independently the efficiency factor (f) of initiator (> 0.6 for diazo initiator), the number of radicals from initiator could also allowed to calculate the livingness of the polymerization reaction. In the case of our work, we expected further the number of radicals from initiator, the transfer coefficient of the chain transfer agent CTA-1 (expected low Ctr) plays a crucial role in the addition fragmentation and influence the amount of radicals generated in the system. Normally, the concentration of active centers should be very low to reduce the probability of terminations reactions in a RAFT polymerization using a CTA with a high Ctr.  For this reason, additional experiments will be performed in order to determine the Ctr of CTA-1 subsequently evaluate its impact on the polymerization RAFT and reported in a subsequent manuscript. Additionally, manuscript was amended, by adding the next information after Figure 5.

Data from the polymerization of MMA carried out at different concentration of CTA-1 clearly demonstrated the impact of [AIBN] in the correlation between the experimental and theoretical molecular weights of PMMA´s. As observed, the SEC traces of PMMA´s (Figure 5) shift towards higher molecular weights with the diminution of the [CTA].  It is well-known that the amount of AIBN can not only predict the number of dead chains into the RAFT polymerization but also the amount of active centers produced during the addition-fragmentation process [32,33]. Notwithstanding, the chemical structure and steric hindrance of CTA-1 could greatly diminish the transfer coefficient (Ctr) of the chain agent and cause a slow re-initiation of leaving radicals R from CTA-1 giving rise to a retardation in the reversible deactivation process.

3) In my opinion the linear fit in Fig. 2A is misleading. After around 10 h the polymerization seems to accelerate as there is more of active centers. In case of Fig. 2 B most of point could be fitted rather by logarithmic fit, which is typical for polymerizations with slow initiation.

Answer: we agree with the reviewer in this slight behavior of acceleration after 10 hours of polymerization, unfortunately at this moment we have not a clear and justified explanation for this minor but still perceptible behavior. On the other hand, as a suggest of other reviewer the dashed line in Figure 2b corresponding to a simple linear fit of the data points was removed and replaced with a line depicting Mn,Th as a function of conversion.

4) I have some comments to Fig. 4. In case of step describing “RAFT equilibrium” the authors wrote in the bracket: “is this stage reversible?” I would like to ask: who should answer this question? Reviewer, readers, or authors? The scheme suggests that reinitiation cannot by realized by leaving R groups, while in the main text it is suggested that some “part of CTA-1 acts for generating oligomeric chains with low monomer units” (line 199 -200). If really there is no reinitiation then we cannot say that this is RAFT mechanism. It is also interesting what is happening with the leaving R radicals. Is it possible recombination of two R radicals?

Answer: we thank the reviewer for its comment. In fact, the Figure 4 was modified and question related to the stage reversible in the RAFT mechanism was removed. As mentioned in the manuscript, the radicals R from the RAFT agent re-initiate the polymerization reaction through growth of polymeric chains. The system reported in this work corresponds to RAFT functionalized polymer which has been confirmed by the formation of copolymers from the chain extension of macroRAFT-PMMA. Due to the chemical structure and steric hindrance, the leaving radicals R from the CTA-1 could cause a slow re-initiation (inhibition in the case of S and BuA). This can probably permit the recombination of two R radicals mostly in the reactions with a higher concentration of CTA-1. On the other hand, PMMA obtained at a concentration [CTA]/[AIBN] =0.25/1 exhibited a good theoretical/experimental molecular weight correlation which indicate that all polymeric chains are RAFT functionalized.

5) Mass ratios of monomer, macro-CTA and AIBN in Table 3 are not informative. I suggest presenting molar ratios.

Answer: [monomer]/[macro-CTA]/[AIBN] mass ratio was replaced in Table 3 (and in the text) by [monomer]/[macro-CTA]/[AIBN] molar ratio as properly suggested by the reviewer.

6) It is quite surprising that polymerization of butyl acrylate was ineffective as the propagation rate constant in case of this monomer is rather high. What could be the explanation of this phenomenon?  

Answer: effectively butyl acrylate presents a high propagation rate constant and even with this, we were not able to go further of an inhibition stage (at the reaction conditions reported in the manuscript). We coincide that the chemical structure and steric hindrance of the leaving radicals R from the CTA-1 could cause inhibition in this monomer. We believe that the conditions studied in this report (excess of CTA-1 in relation with the AIBN), plus the structural nature of the monomer (secondary radical propagating instead of a tertiary radical propagating for MMA) had influence on the prolonged inhibition behavior observed for this monomer. It is important to mention that no other reaction conditions have been tested for butyl acrylate polymerizations in presence of CTA-1. We will prove with other conditions and also with an asymmetric trithiocarbonate containing 1 diphenylmethyl group in one side and one butyl group on the other side expecting to answer in a better way this and other questions for oncoming papers.

The reviewer is thanked for its valuable comments and corrections.

Round 2

Reviewer 1 Report

The authors have sufficiently addressed my previous concerns, and I recommend publication of this manuscript.

Reviewer 2 Report

I am satisfied with the response to my comments and recommend to accept the manuscript for publication.